# A Model for Direct Effect of Graphene on Mechanical Property of Al Matrix Composite

Hongshuo Sun [1], Na Li [2], Yongchao Zhu [1,*] and Kailiang Liu [1]

[1] Department of Railway Engineering, Zhengzhou Railway Vocational and Technical College, Zhengzhou 450001, China; 11082@zzrvtc.edu.cn (H.S.); liukailiang@zzrvtc.edu.cn (K.L.)
[2] School of Mechanics and Safety Engineering, Zhengzhou University, Zhengzhou 450001, China; zzulina@zzu.edu.cn
* Correspondence: zhuyongchao@zzrvtc.edu.cn

**Abstract:** Direct effect of graphene on mechanical property of Al matrix composite has been studied by using molecular dynamic (MD) methods. The models of graphene-reinforced composite are achieved by modeling the sintering system consisting of Al particles and graphene nanosheets (GNSs), while pure Al models are obtained by deleting graphene in the composites. Structural analysis on composites indicate the increment of GNSs can promote the densification of metal matrix, increase the porosity in composite, and restrict the metal grain size. Such analysis is also performed on pure Al models, and the similarity in structure between pure Al and composite models is confirmed by the tiny difference in the nanopores, atomic images, and the number of ordered atoms. Tensile processes on the similar structures with or without graphene reveal that the direct effect of graphene shows an obvious anisotropy, low graphene content may weaken the composite in some directions, while high graphene content can strengthen the composite in more directions. However, the highest content of GNSs just brings a slight increase of 2.7% in tensile strength. The atomic images of crack propagation and the atomic stress confirm that graphene is not efficient in load transfer. Therefore, the direct effect of graphene is believed to play a very small role in strengthening mechanisms.

**Keywords:** molecular dynamic; graphene; composite; direct effect; strengthening mechanism





## 1. Introduction

As is well-known to all, metal matrix composites (MMCs) can be improved by various reinforcements, and a large amount of efforts have been made to characterize the mechanical behavior of particle reinforced MMCs. The strengthening mechanisms may be divided into direct and indirect categories. Direct strengthening is derived from the mechanism for the behavior of continuous fiber-reinforced composite, while indirect strengthening results from the thermal mismatch between metallic matrix and a high stiffness ceramic particle [1]. However, the theories based on reinforcements of fiber and ceramic particle may not be quite suitable for two-dimensional material such as graphene, which has is considered to be more effective than conventional reinforcements.

In the studies of direct strengthening of graphene, the theory based on fiber-reinforced composite is applied frequently, where graphene is taken as a continuous rectangular platelet. Tang et al. reported that the addition of only 1.0 vol.% GNSs lead to a 94% improvement in yield strength (268 MPa) of GNS-Ni/Cu composites, which agrees well with the modified shear-lag model [2]. Chu et al. found a remarkable increase in yield strength of graphene-copper composites at 8 vol.% GNP content, which is below the theoretical value of the Halpin-Tsai model [3]. Shin et al. provided a new model to predict strength of MMCs, which is verified by C/Ti and C/Al composites [4]. Although the measured tensile strength values are largely coincided with the theory, the indirect effect cannot be excluded in experimental studies. Molecular dynamic (MD) models have been employed to investigate the tensile behavior of graphene-reinforced composite too, where graphene

nanosheet (GNS) is embedded in metal matrix with an unfolded state. Duan et al. built an MD model of graphene-embedded copper to investigate the effects of chirality and number of graphene layers on mechanical properties [5]. Zhu et al. employed a MD model of graphene/Al laminated composites, where graphene blocks propagation of dislocations and bears most of the loads [6]. However, the actual state of GNS in metal matrix is very complicated. Because graphene is prone to bending, load transfer may not be as efficient as in the case of plate state. Even if the strengthening effect is explored with a simple model [7], the indirect effect of graphene cannot be avoided. Furthermore, the direct effect of reinforcements seems to have no heavy influence on mechanical property, based on only a slight rise of 4.4% of tensile strength in TiC particles reinforced Al2219, where Krajewski et al. used a thermomechanical treatment to provide a homogeneous distribution of dislocations in both composite and the unreinforced alloy [8].

In the extent of indirect strengthening of graphene, it is hard to quantify the contribution in composite. Nevertheless, grain-size dependent mechanical behavior of metal has been well-studied, and the grain size is consistent with graphene content. Meysam et al. argued that the strengthening of nanocomposites reinforced by GNPs would be primarily controlled by the indirect impact of grain boundary pinning from GNPs, proposing a developed equation of the Hall–Petch relation [9]. Choi et al. investigated strengthening efficiency of aluminum-based composites with grain sizes ranging from 250 nm to 65 nm, and the increment of the flow stress is in accordance with the values calculated from the Hall–Petch relationship [10]. Shin et al. analyzed the reduction in grain size with nearly 46% improvement in the ultimate tensile strength noticed, compared with pure aluminum [11]. However, the direct effect of reinforcements was not evaluated in these studies as well.

In summary, there are no studies focusing on the direct effect of graphene on mechanical property only, which can avoid indirect effects at the same time. In this study, Al/graphene structures are achieved from designed models for powder sintering with different graphene content, and pure Al bulks similar to the composite structure are obtained by deleting graphene in composite, structural analysis is performed to verify the similarity. Thus, the direct effect of graphene can be analyzed by conducting the same tensile simulations on the similar structures with or without graphene.

## 2. Model and Method

To gain reasonable models for studying the effect of reinforcements, composite structures close to the fact are built on the basis of the experimental process of powder metallurgy. As shown in Figure 1, the initial model similar to a simple cubic consists of 27 Al spherical nanoparticles with a diameter of 48 Å, and the spaces between Al particles are filled by a certain amount of square GNSs with a side length of nearly 24 Å. Based on the experimental review, the average sizes of metal particles ranges from 25 to 75 μm, while the mean platelet diameter of GNSs fluctuates from 5 to 15 μm [12–16]. Therefore, the diameter of Al particles here is a little larger than the size of GNSs. It should be noted that the crystal orientation of Al particles are random, and the adjacent Al particles keep 8.1 Å away from each other. The empty space is bigger than similar studies of Al/SiC models, where spherical reinforcing particles can fill in smaller space [17,18]. Moreover, the angle of rotation around the center of GNSs are random as well, so it is necessary to leave a larger space for insertions of GNSs. The five structures containing a different number (1, 8, 14, 20, and 27) of GNSs are correspondent to the weight fraction of graphene from 0.1% to 2.4%. The detailed structures of composites with highest and lowest content of graphene are displayed in Figure 1. Because periodic boundary conditions are applied in the x, y, and z directions, a few GNSs are displayed incompletely. The number of atoms in every Al particle is 3565, and the number of carbon atoms in a piece of GNS is 190. Compared with most MD models in literature where Al matrix and one or several GNSs form a sandwich-like structure [19–21], the composite system in this paper is more realistic.

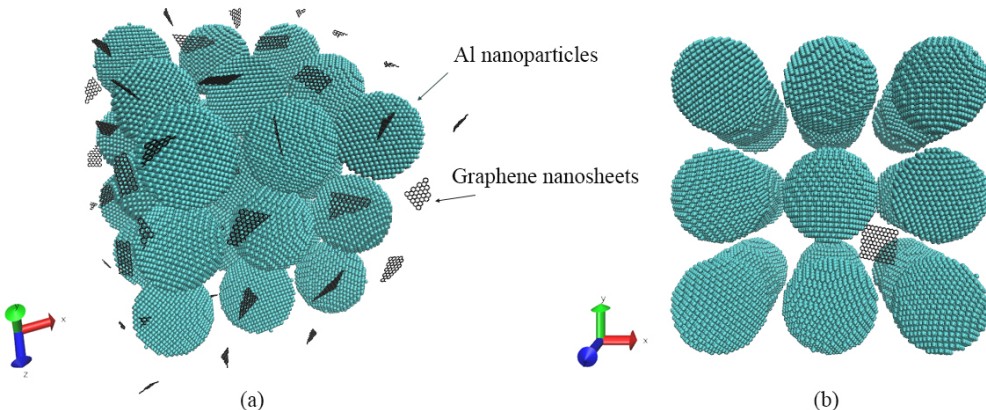

**Figure 1.** Schematic of the initial model for composite systems containing (**a**) 27 GNSs and (**b**) 1 GNS.

To gain a compact structure for sintering, the initial models are relaxed in isothermal-isobaric (NPT) ensemble at 300 K and 1 atm for 100 ps. As a result, the Al particles and GNSs form a whole, resulting from the external press and surface energy. Then, the relaxed models are sintered in NPT ensemble as well. At first, the temperature increases from 300 K to 773 K in 300 ps, with the external press rising from 1 atm to 500 atm; secondly, the entire system maintains such temperature and pressure in the next 300 ps; finally, the temperature and pressure fall back to 300 K and 1 atm, completing the sintering process. This process is designed according to experimental studies on spark plasma sintering by our group [22], which fabricated graphene nanoplatelets-reinforced 7075 aluminum alloy composite.

The mechanical property of sintered composite is investigated by modeling the quasi-static tension, where the shape of simulation box is changed in every 100 timesteps, with an engineer strain rate of 0.001/ps. The tensile process is conducted in the microcanonical (NVE) ensemble, with the temperature controlled at 300 K by explicitly rescaling the velocities. Such a process has been applied in MD studies on the mechanical properties of graphene/Al composite [6,23]. Here, the entire system can reach equilibrium quickly under a low engineer strain rate at every stage.

To explore the direct effect of graphene, GNSs in the sintered composites above are deleted, and the remaining structure is relaxed in NPT ensemble at 300 K and 1 atm for 100 ps, to relieve the inner stress. Thus, a pure Al structure similar to Al/graphene composite can be obtained, and the same tensile process is operated again. If GNS is deleted before sintering, the pure Al structure is comprised of bigger grains, and the inner structure such as grain boundary and dislocation is quite different from composite. Thus, the direct effect is overlaid by indirect effect, which cannot be studied separately. It is worth noting that there are no visible empty spaces after deleting, and carbon atoms just filled the gaps between Al atoms. It means that there is no need to fill in extra Al atoms.

All the modeling and analysis processes in MD simulations are operated by the LAMMPS code, and the visualization is conducted by VMD software. In all MD models, periodic boundary conditions are applied in three directions, and the time step is set to 1.0 femtosecond. A stable file of *eam* potential is used to describe the interaction between Al atoms [24], and the *airebo* potential is applied to the interatomic interaction in GNSs [25]. Al-C interaction generally takes the style of *morse* potential, which is expected to be more suitable than *Lj* potential (a classic pair potential) for modelling the interface under cohesive zone law [26].

## 3. Results and Discussion

### 3.1. Comparisons on the Inner Structures

The sintering process of metal-graphene system has been investigated deeply in simple MD models, which are built with only one sheet of graphene and several metal particles [23,27]. Here, the Al/graphene composite are achieved with a different number of GNSs in a larger scale. The details in powder metallurgy such as plastic deformation

and mass flow can be observed similarly in Figure 2. This densification process can be monitored by the change in volume and morphology. Because there is a large space for the insertion of graphene, the initial system volumes after relaxation are different. Since the particles are not close to each other, the system volumes drop sharply at first in the heating stage, followed by an increase owing to the thermal expansion. Then, the stable pressure and temperature compresses the structures slowly, and the final structures are obtained after cold shrink in the cooling stage.

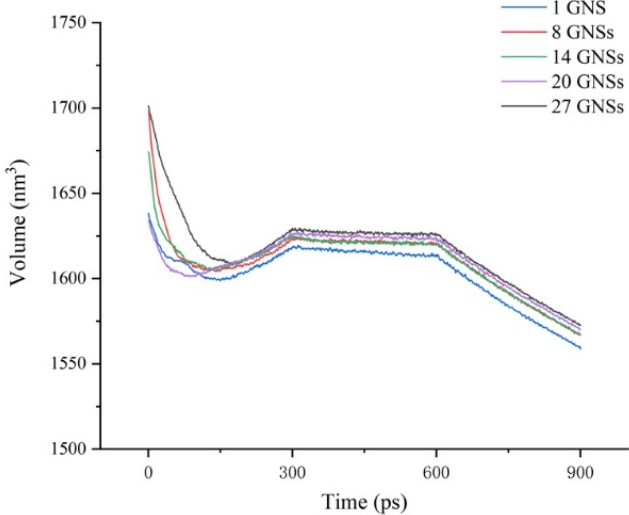

**Figure 2.** The evolution of system volume and atomic configurations in the sintering models.

The volume fraction of Al atoms in a composite can be easily figured out in MD simulations to explore the effect of graphene on inner structures. As reported, graphene can promote the densification of metal matrix composite [28]. So, it can be confirmed by that Al volume in composites decreases from 1558 nm$^3$ to 1544 nm$^3$ in Figure 3a, with the growth in the number of GNSs, while the total volume of composite increases from 1559 nm$^3$ to 1573 nm$^3$. In the scale of this study, the interspaces between atoms bigger than a spherical diameter of 4 Å is defined as pores. However, the porosity of composite visibly reveals that more GNSs give rise to formation of nanopores, and the total volume of nanopores increase from 0.248 nm$^3$ to 0.704 nm$^3$, according to the images and trends in Figure 3b. There is no contradiction here, because the volume of nanopores beside GNS is partly regarded as the Al volume, which is computed by calculating the Voronoi tessellation of each atom. For further analysis of the structure, centro-symmetry parameter (CSP) analysis is performed on sintered composites. The local lattice disorder can be clearly displayed in CSP values from 0 (perfect lattice) to 2.3 (surface atoms). That is to say, blue areas stand for Al atoms with perfect lattice, white areas represent defects such as grain boundaries, and red atoms imply nanopores. In spite of the random in crystal orientation of initial Al particles, Al grains with large sizes are formed in the sintering process, since several big blue areas can be observed in Figure 3. Obviously, the size of Al grains reduce with the rise in the number of GNSs (Figure 3c). It should be noted that the GNSs are located at the Al grain boundaries in all the models, which is consistent with metal matrix composite prepared by experimental methods [11,29]. At the interface between Al and GNS, Al atoms on both sides of graphene are expected to be organized in the {111} facet of the face-centered cubic (f.c.c.), facilitating the crystallization of metal atoms [27–30]. So, Al atoms beside GNSs may be considered as the hexagonal close-packed (h.c.p.) order. However, the lattice parameter for the close-packed plane of Al grain is a little bigger than the size of the honeycomb lattice of graphene, and GNSs in composite almost present the blending state. Thus, it is impossible for Al atoms beside graphene to form perfect lattice. With the growth in graphene content, more Al atoms affected by graphene lead to the extension in grain boundaries and the decrease in the size of Al grains. Common neighbor analysis (CNA) is operated to count

the number of atoms arranged with f.c.c. or h.c.p. order, and both of them are classified into ordered atoms, because f.c.c. atoms can be considered as atoms with perfect lattice, h.c.p. atoms may be considered as stack fault. Since the total number of Al atoms in every model is 96,255 (=3565 × 27), the percentage of order atoms can be calculated easily. As expected in Figure 3d, the composite sintered form the same system with less GNSs has more f.c.c., h.c.p., and ordered atoms, and the percentage of ordered atoms drops from 81.5% to 45.7% with the increment of GNSs. Nevertheless, the change in the number of atoms organized in the h.c.p. order is tiny, which means that Al atoms arranged in close-packed plane on both sides of graphene has not formed h.c.p. structures eventually. Therefore, it is concluded that graphene can restrict the metal grain size rather than facilitate metal crystallization.

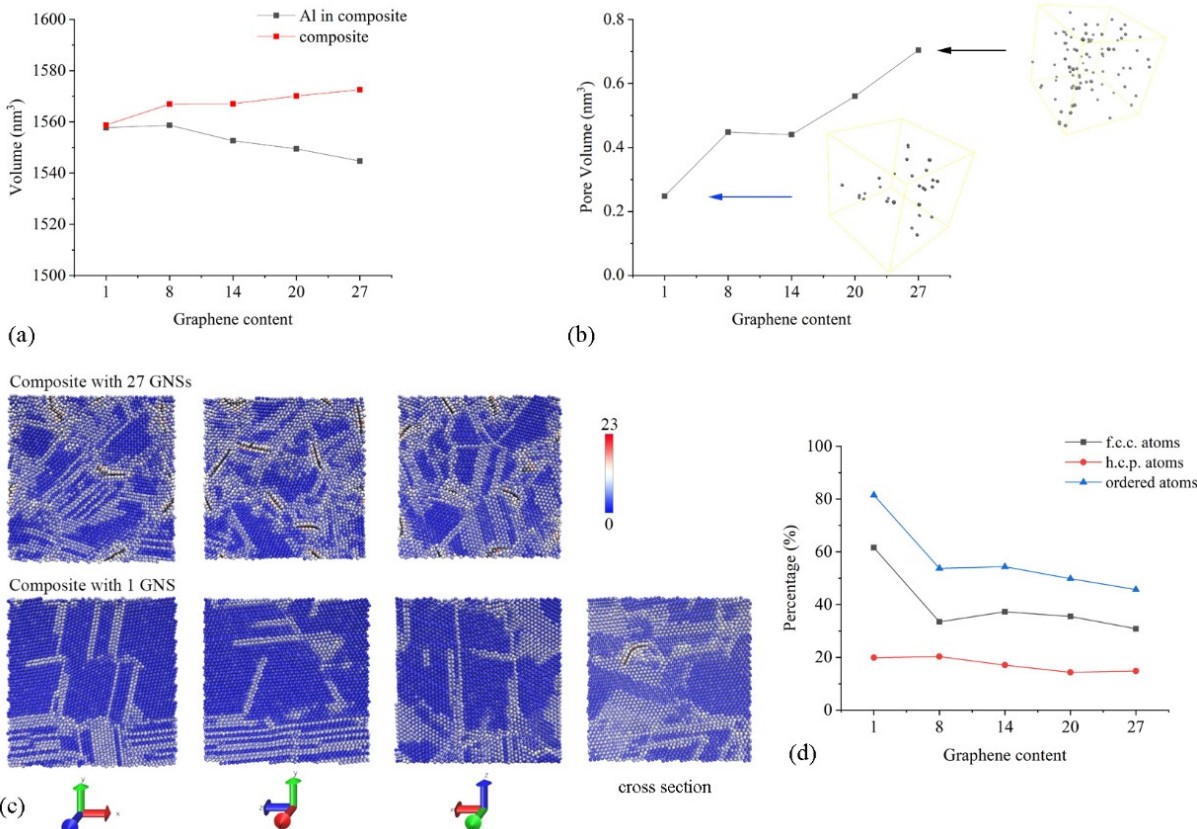

**Figure 3.** The structure analysis on sintered composite. (**a**) The volume of Al part; (**b**) the porosity; (**c**) CSP images; (**d**) and CNA curves.

For studying the direct effect of graphene, pure Al structures similar to the composites are built through deleting graphene in composites, because the GNSs only locate at metal grain boundaries as stated above. After relaxation in NPT ensemble at 300 K and 1 atm, Al atoms that were around GNSs previously can rearrange spontaneously with the effect of surface energy. Of course, they do not move vigorously. Thus, the structure similar to the composite can survive. Structural analysis can account for this similarity. With removal of graphene, Al atoms beside graphene become closer to each other, the total volume drops a little, which ranges from 1559 $nm^3$ to 1566 $nm^3$ (Figure 4a). Meanwhile, a few pores diminish (Figure 4b), leading to a very small drop in the porosity which ranges from 0.200 $nm^3$ to 0.648 $nm^3$. Figure 4c illustrates that there is no obvious gap in the structures of Al grains after removing graphene, compared with the CSP images in Figure 3c. The location and crystal orientation of Al grains in pure Al bulk are the same as that in composites. Only the grain boundaries that originally contained GNSs look to be narrowed slightly. CNA values in Figure 4d also indicate that the percentage of Al

atoms arranged in perfect lattice increases a little, which reduces from 82.1% to 54.9%. The more GNSs in original composites, the more Al atoms turn into f.c.c. structure. To some extent, the grain refinement of graphene can be considered as increasing both the number and width of grain boundaries. In general, Figure 4 indicates that pure Al obtained by deleting GNSs are a little more compact and have less defect atoms than composite, but the similarity in structure keep the study of mechanical property from the indirect effect of graphene. Therefore, it is significant for the study of direct mechanical behaviors on the basis of such models.

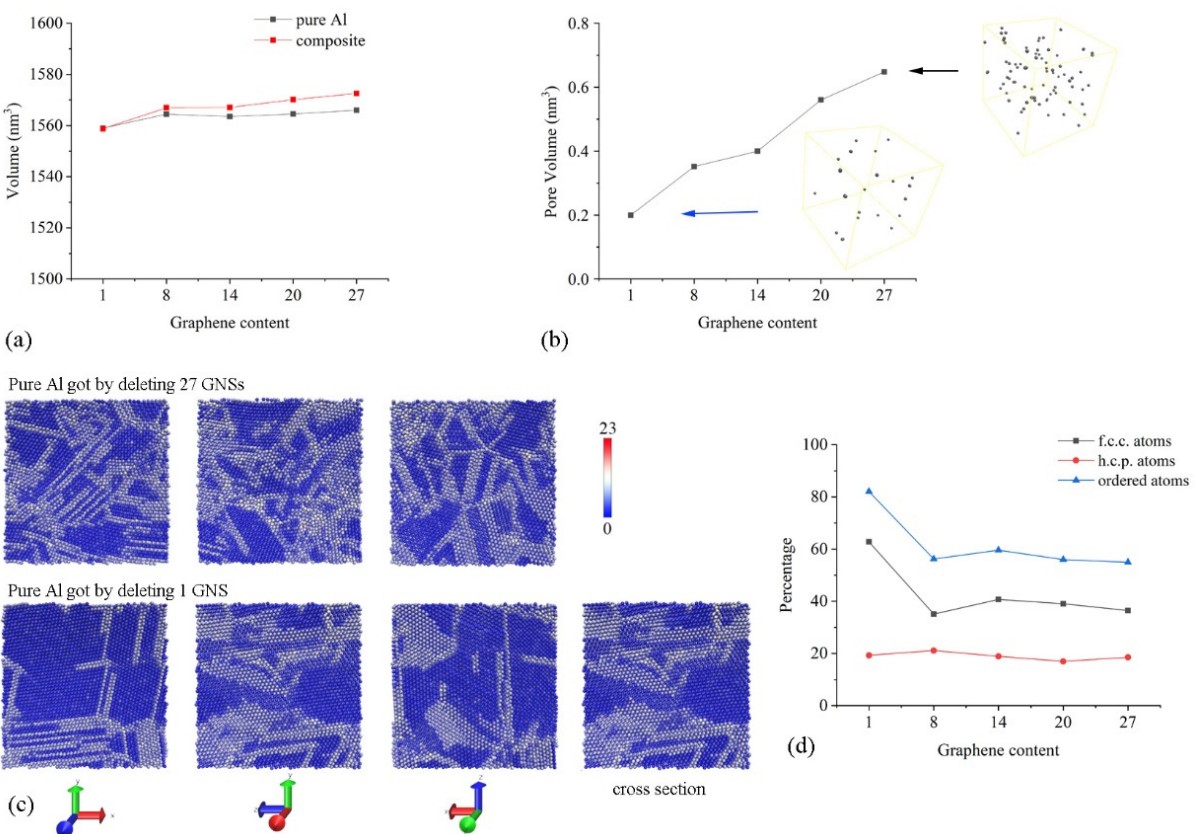

**Figure 4.** The structure analysis on pure Al bulk obtained by deleting GNSs. (**a**) The volume of Al part; (**b**) the porosity; (**c**) CSP images; (**d**) and CNA curves.

### 3.2. Comparisons on the Tensile Processes

To verify the direct strengthening in graphene-reinforced composite, tensile simulations are conducted on the designed models above. In view of the two-dimensional properties and random orientation of graphene, both the composite structures and unreinforced structures are stretched in three directions, and the total tensile strength is defined as the sum of tensile strength for the three directions. As shown in Figure 5a, the total tensile strength of pure Al structures decreases from 20.9 GPa to 19.8 GPa, that is in line with inverse hall-patch relation which brings a drop with the reduction in grain size [31]. The total tensile strength of composite rises from 19.7 GPa to 20.4 GPa, implying that graphene is helpful for the improvement of mechanical property. However, composite containing less GNSs seems to have no advantage on tensile strength, and even the composite with only one GNS becomes softer obviously in two directions. This means that the direct strengthening of graphene may have a negative effect on mechanical property when graphene content is low. When the number of GNSs increase to 14, the composites become stronger at a certain direction, suggesting that the tensile property is in relation to the orientation of GNSs. After the number of GNSs exceeds 20, the tensile property of composite has been improved thoroughly, but there is a clear gap between the increments in different directions.

As a whole, Figure 5b reveals that the tensile strength is raised only 2.7% on average (total about 0.5 GPa for three directions), even with the highest graphene content. The structures containing 27 GNSs here is equivalent to the composite with 2.4 wt.% graphene experimentally. Whereas the tensile strength can be improved 45% at least in experiments with only 0.54 wt.% graphene [2–4], where grain refinement may play a more important part. Thus, it can be concluded that the direct strengthening of graphene is much lower than the indirect effect.

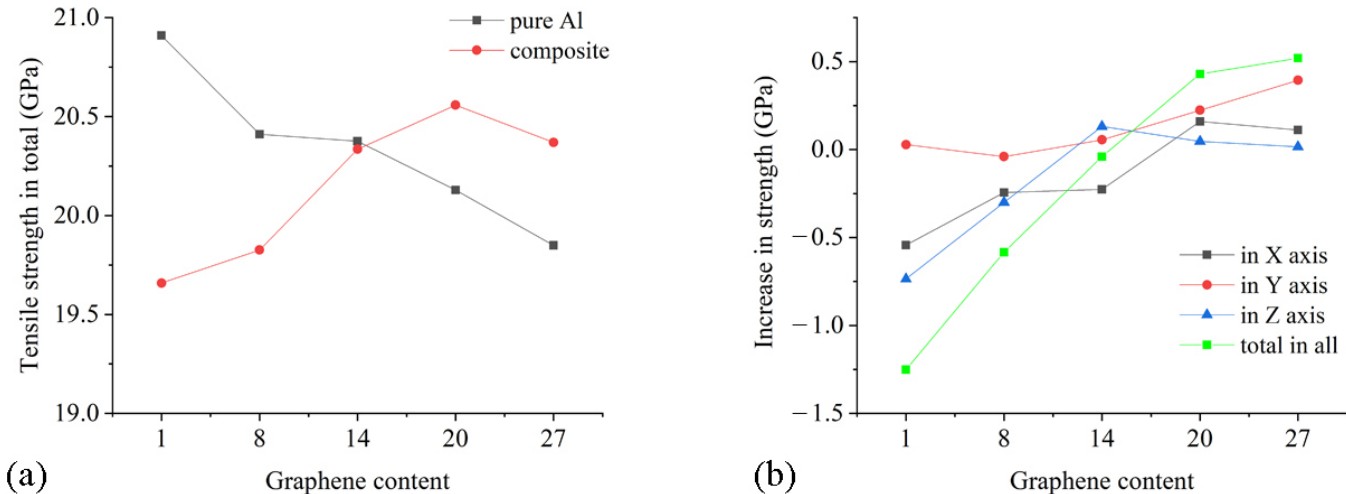

**Figure 5.** (**a**) The tensile strength in total and (**b**) increased strength as a function of graphene content.

In Figure 6, the CSP images under different strains present the changes in inner structure in the tensile process, and the models with 1 and 27 GNS are selected to observe the direct strengthening behavior of graphene. As commonly believed, the dislocation in metal grains moves under tensile deformation, eventually absorbed by grain boundaries [15,32]. If graphene content is low, most grain boundaries are not reinforced by graphene. When stretched in the *y*-axis, the initial crack in the composite is formed at the same location in pure Al bulk with a similar structure, and the crack propagates the path is far away from graphene (Figure 6a,b). Thus, the almost equal values of tensile strength are measured (Figure 6e). When stretched in *x*-axis, GNS seems to hinder the plastic flow of adjacent Al atoms, leading to stress concentration that accelerate the formation of cracks. Thus, the tensile strength of pure Al is bigger than that of composites combined with larger elongation at failure. If graphene content is high, most grain boundaries are strengthened by graphene. The initial crack in composite may originate from the same void too, as shown in Figure 6c,d, and there is no visible gap between the tensile strength of pure Al and composite (Figure 6f), when stretched in the *x*-axis. However, a great number of GNSs can give rise to global enhancement in structure, leading to a different route for crack extending, so the tensile property of composite has been largely improved in the *y*-axis. To sum, the direct effect of graphene shows an obvious anisotropy, low graphene content may weaken the composite in some directions, while high graphene content can strengthen the composite in more directions.

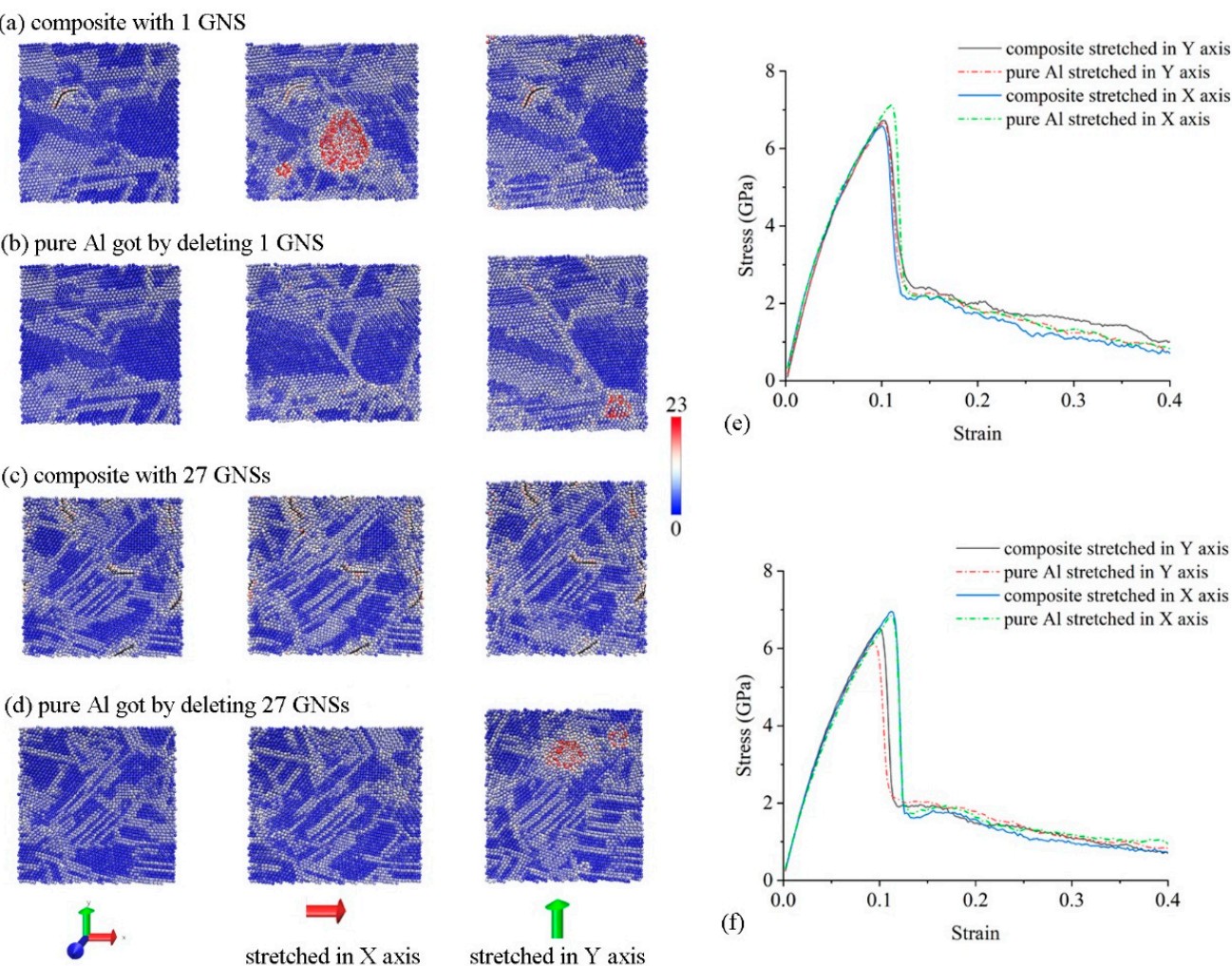

**Figure 6.** (**a**–**d**) The atomic details of crack propagation at maximum stress in tensile models containing 1 and 27 GNSs, and the stress–strain curve of models containing (**e**) 27 GNSs and (**f**) 1 GNS.

The stress distribution of Al atoms and GNSs is tinted in Figure 7 to reveal the direct effect of graphene. Since the negative value of per-atom stress stands for tensile stress, Figure 7a reveals that Al atoms beside graphene bear compressive stress, owing to the lattice mismatches when Al atoms are bonded to the graphene lattice. No matter in the composite with less or high graphene content, only a small area in every GNS bears tensile stress. There is no larger increase in both the tensile area and stress value of carbon atoms when loaded to the maximum stress as shown in Figure 7b–d. Furthermore, a great rise in the tensile stress of most Al atoms can be found, no matter in the composite with 27 GNSs or 1 GNS. That is, the load may not be transferred effectively by graphene of such sizes in these models. Therefore, we conclude that graphene is likely to introduce only a small impact on the mechanical property of the composite, regardless of the indirect effect. The method for improving graphene-reinforced MMC can be paved through raising the efficiency of load transfer in the next stage.

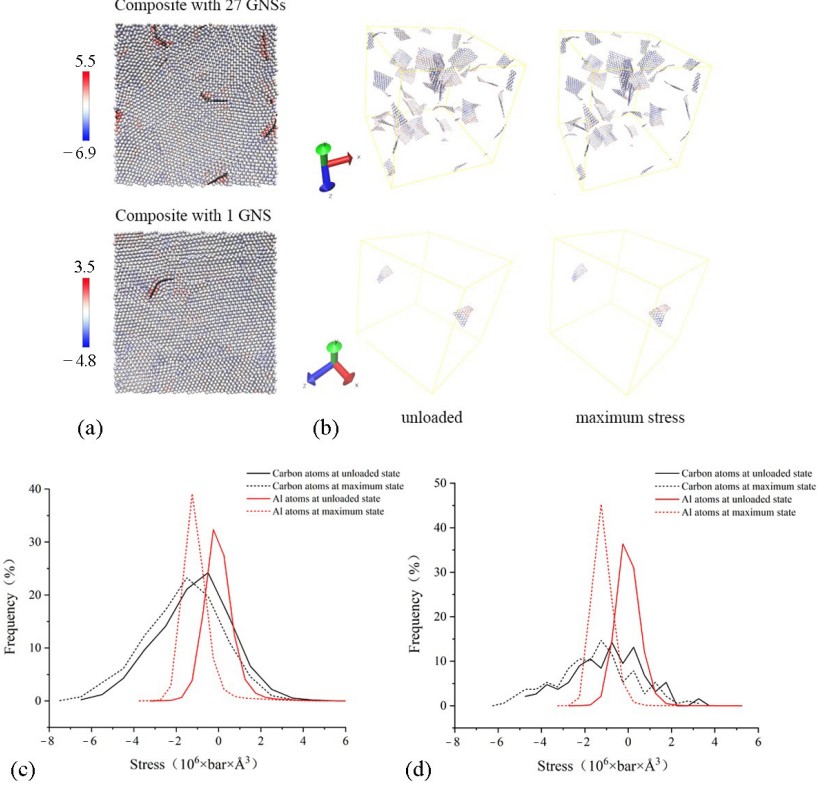

**Figure 7.** The atomic stress of Al atoms and GNSs. The images of stress distribution in (**a**) Al atoms and (**b**) GNSs, (**c**) the stress distribution of atoms in composite with 27 GNSs, and (**d**) the stress distribution of atoms in composite with 1 GNS.

## 4. Conclusions

The direct effect of graphene in composites is investigated by employing MD simulations. The Al/graphene composite structures close to the fact are produced by modeling the powder metallurgy process, and then pure Al structures similar to the composites are obtained by deleting the graphene in composites. The difference in structures are characterized by computing the metal volume, the porosity, and the grain size. With an increase on graphene content, the total volume of composite increases from 1559 nm$^3$ to 1573 nm$^3$, the Al volume in the composite decrease from 1558 nm$^3$ to 1544 nm$^3$, the total volume of nanopores increase from 0.248 nm$^3$ to 0.704 nm$^3$, and the percentage of ordered Al atoms reduce from 81.5% to 45.7%. Such structural analysis is also performed on the pure Al models obtained by deleting GNSs in composites, and the results reveal that there is no great distinction between the pure Al and composite models. The total volume drops a little, which ranges from 1559 nm$^3$ to 1566 nm$^3$; a few pores diminish, leading to a very small drop in the porosity, which ranges from 0.200 nm$^3$ to 0.648 nm$^3$; the percentage of ordered Al atoms increases a little, which reduces from 82.1% to 54.9%. Therefore, the tensile process can be conducted on both the composite and pure Al models to reveal the direct effect of graphene. The tensile strength reveals that the composite containing less GNSs become softer than pure Al models in some directions, while the composite containing more GNSs can become stronger overall. However, even if the number of GNSs grow to the maximum that is beyond the content in experiments, only a slight rise of 2.7% can be brought in tensile strength. In the light of the crack propagation path, the stress–strain curve, and the distribution of stress, it can be concluded that graphene has not served as an efficient role of load transfer. Therefore, direct strengthening of graphene is believed to have little impact on the mechanical property of MMC.

**Author Contributions:** Writing—original draft, H.S.; Writing—review and editing, N.L.; Conceptualization and data curation, Y.Z.; Investigation and formal analysis, K.L. All authors have read and agreed to the published version of the manuscript.

**Funding:** This research was funded by [the Program for Natural Science Foundation of Henan Province] grant number [232300421344].

**Institutional Review Board Statement:** Not applicable.

**Informed Consent Statement:** Not applicable.

**Data Availability Statement:** Not applicable.

**Conflicts of Interest:** The authors declare no conflict of interest.

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
