# Peer review of "A Model for Direct Effect of Graphene on Mechanical Property of Al Matrix Composite"

_metals, doi:10.3390/met13081351_

Round 1
Reviewer 1 Report
Overall, this manuscript conclude that direct effect of graphene play very small role in the strengthening mechanism. This conclusion based on results from numerical simulation only. Disclaimer or scope need to put here.
1) Literature review need to add table of review for experimental results that obtain for mechanical properties. Researchers reported graphene able to improve mechanical properties (experimentally). After extensive literature review, author able make proper conclusion and comparison between numerical and experimental value.
2) Method for pure AI matrix need to clarify, current only by deleted GNS. Thus, the empty space by GNS should fill with Al. Figure 1, Al matrix represent by Al nanoparticles. In experimental, it should in matrix by after sintering. Need to verify about this.
3) results & Discussion : Figure 5, should have comparison with experimental results from literature review. Thus, researchers reported that Graphene able to increase mechanical properties. How to explain in theory that pure AI reduce tensile strength by increase graphene content? what mean pure Al here (isn't without GNS)
$) Conclusion : after improvement of literature review and results, more concrete and objective conclusion can be make.
minor english editing is required.
Reviewer 2 Report
The manuscript by H. Sun et al. addresses the mechanical reinforcement of Al composites by adding different quantities of graphene nanosheets to Al before sintering. The authors employ molecular dynamics simulations to this end. First of all, the paper should not be published without a proper language revision, since this is, in my view, one of its most serious flaws. The originality and importance of the results presented here are, at best, medium, since modeling graphene sheets embedded in Al nanocomposites by MD is a much-explored theme. The simulations setup is somewhat interesting since modeling the sintering process instead of modeling full melting and recrystallization for nanocomposite reinforcement is not so common. The paper is concise and well organized. I would support the publication of this paper if the authors address the following observations:
a) Introduction section: The following sentence in the last paragraph of introduction is not entirely clear “In a word, there is no special discussions of direct effect of graphene on mechanical property” What do you mean by special discussions?
b) Model and method section: In Fig 1, the GNS do not appear square. What are their exact dimensions and number of atoms and why did you chose these dimensions? Is this Figure correspondent to 27 GNS (2.4% w/w)? This should be clear in the paper. Al particles are 2A apart, but what is the distance between Al atoms and the GNS introduced?
c) Model and method section: The autors state that “At first, the temperature increases from 300 K to 773 K in 300 ps, with the external press(ure) rising from 1 atm to 500 atm; secondly, the entire system maintains such temperature and pressure in the next 300 ps; finally, the temperature and pressure will fall back to 300 K and 1 atm, completing the sintering process.” What is the basis for this sintering method in MD? Was it validated before? If yes, cite references.
d) Model and method section: Provide references also for the method of obtaining the mechanical properties, particularly for the strain rate and emsemble used. The explanation of this method lacks detail and/or references.
e) Model and method section: Deleting the GNS from the system after sintering introduces defects and does not equal the structure of pure Al you would want to compare it with. Shouldn’t the authors delete the GNS before sintering? This is one of my main doubts, what would be the result in terms of mechanical properties if you deleted the GNS before sintering? Please discuss this with results.
f) Results and Discussion section: In Figure 2 the pictures of the square section of the simulation box are not clear, nor understandable nor useful, I suggest the authors to remove them.
g) Results and Discussion section: Figure 3 can be improved. In Figure 3a) and 3b) what are the units of graphene content in the legend of x axis? I know it’s the number of GNS, but that should be explicit. In y axis, volumes should have the same units. In Figure 3 c) what are the units of the side scale 0 to 2.3? In figure 3 d) I suggest substituting the number of Al atoms by % of Al atoms (in FCC -face cubic centred (not ffc as is in the Figure) or HCP or ordered (what does ordered means in his context?)) relative to total number of Al atoms.
h) Results and Discussion section: The authors state that “With removal of graphene, Al atoms beside graphene rearrange in the relaxation, the total volume drops a little (Fig. 4a), just a few pores diminish (Fig. 4b), leading to a very small drop in the porosity which range from 200 AÌŠ3 to 648 AÌŠ3 . This rearrangement is done at 300K, but it is not clear how it is done? By energy minimization? At 300K Al structures do not rearrange spontaneously. Again, the authors should compare their results with sintered pure Al spheres, that is, removing GNS before sintering stage, not after.
i) Results and Discussion section: The authors state that “In general, it is still significant for the study of direct mechanical behaviors based on the similar structure between composite and pure metal bulk. “ Please rephrase it because it not clear what you mean.
j) How do you define total tensile strength, is it an average of tensile strength for the three directions? Please explain this in the paper.
k) Results and Discussion section: Figure 6 lacks the legend for CSP atom types, similar to that in Figure 4.
l) Results and Discussion section: It would be helpful for understanding more clearly the mechanical deformation of the composite if the authors performed a study of dislocations.
m) Results and Discussion section: In Figure 7 b), the simulation box boundaries are not clear. Figure 7 c) and d) are not comprehensible, what do they indicate? Atomic number refers to a very well defined property of different elements, and has nothing to do with what the authors refer here. I suppose, you are referring to the index number of carbon or Al atoms in the structures, but unless we know how the numbering was done, this does not make sense. Please improve it.
The paper should not be published without a proper language revision, since this is, in my view, one of its most serious flaws
Round 2
Reviewer 2 Report
The authors have fully addressed my suggestions for improving the paper, therefore i support its publication